# Distributed Representations of Words and Phrases and their Compositionality

**Tomas Mikolov**
Google Inc.
Mountain View
mikolov@google.com

**Ilya Sutskever**
Google Inc.
Mountain View
ilyasu@google.com

**Kai Chen**
Google Inc.
Mountain View
kai@google.com

**Greg Corrado**
Google Inc.
Mountain View
gcorrado@google.com

**Jeffrey Dean**
Google Inc.
Mountain View
jeff@google.com

## Abstract

The recently introduced continuous Skip-gram model is an efficient method for learning high-quality distributed vector representations that capture a large number of precise syntactic and semantic word relationships. In this paper we present several extensions that improve both the quality of the vectors and the training speed. By subsampling of the frequent words we obtain significant speedup and also learn more regular word representations. We also describe a simple alternative to the hierarchical softmax called negative sampling.

An inherent limitation of word representations is their indifference to word order and their inability to represent idiomatic phrases. For example, the meanings of "Canada" and "Air" cannot be easily combined to obtain "Air Canada". Motivated by this example, we present a simple method for finding phrases in text, and show that learning good vector representations for millions of phrases is possible.

## 1 Introduction

Distributed representations of words in a vector space help learning algorithms to achieve better performance in natural language processing tasks by grouping similar words. One of the earliest use of word representations dates back to 1986 due to Rumelhart, Hinton, and Williams [13]. This idea has since been applied to statistical language modeling with considerable success [1]. The follow up work includes applications to automatic speech recognition and machine translation [14, 7], and a wide range of NLP tasks [2, 20, 15, 3, 18, 19, 9].

Recently, Mikolov et al. [8] introduced the Skip-gram model, an efficient method for learning high-quality vector representations of words from large amounts of unstructured text data. Unlike most of the previously used neural network architectures for learning word vectors, training of the Skip-gram model (see Figure 1) does not involve dense matrix multiplications. This makes the training extremely efficient: an optimized single-machine implementation can train on more than 100 billion words in one day.

The word representations computed using neural networks are very interesting because the learned vectors explicitly encode many linguistic regularities and patterns. Somewhat surprisingly, many of these patterns can be represented as linear translations. For example, the result of a vector calculation vec("Madrid") - vec("Spain") + vec("France") is closer to vec("Paris") than to any other word vector [9, 8].

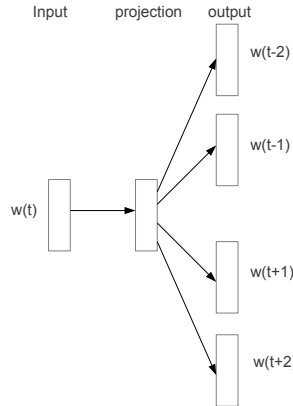

Figure 1: The Skip-gram model architecture. The training objective is to learn word vector representations that are good at predicting the nearby words.

In this paper we present several extensions of the original Skip-gram model. We show that sub-sampling of frequent words during training results in a significant speedup (around 2x - 10x), and improves accuracy of the representations of less frequent words. In addition, we present a simplified variant of Noise Contrastive Estimation (NCE) [4] for training the Skip-gram model that results in faster training and better vector representations for frequent words, compared to more complex hierarchical softmax that was used in the prior work [8].

Word representations are limited by their inability to represent idiomatic phrases that are not compositions of the individual words. For example, "Boston Globe" is a newspaper, and so it is not a natural combination of the meanings of "Boston" and "Globe". Therefore, using vectors to represent the whole phrases makes the Skip-gram model considerably more expressive. Other techniques that aim to represent meaning of sentences by composing the word vectors, such as the recursive autoencoders [15], would also benefit from using phrase vectors instead of the word vectors.

The extension from word based to phrase based models is relatively simple. First we identify a large number of phrases using a data-driven approach, and then we treat the phrases as individual tokens during the training. To evaluate the quality of the phrase vectors, we developed a test set of analogical reasoning tasks that contains both words and phrases. A typical analogy pair from our test set is "Montreal":"Montreal Canadiens"::"Toronto":"Toronto Maple Leafs". It is considered to have been answered correctly if the nearest representation to vec("Montreal Canadiens") - vec("Montreal") + vec("Toronto") is vec("Toronto Maple Leafs").

Finally, we describe another interesting property of the Skip-gram model. We found that simple vector addition can often produce meaningful results. For example, vec("Russia") + vec("river") is close to vec("Volga River"), and vec("Germany") + vec("capital") is close to vec("Berlin"). This compositionality suggests that a non-obvious degree of language understanding can be obtained by using basic mathematical operations on the word vector representations.

## 2 The Skip-gram Model

The training objective of the Skip-gram model is to find word representations that are useful for predicting the surrounding words in a sentence or a document. More formally, given a sequence of training words $w_1, w_2, w_3, \ldots, w_T$, the objective of the Skip-gram model is to maximize the average log probability

$$\frac{1}{T} \sum_{t=1}^{T} \sum_{-c \leq j \leq c, j \neq 0} \log p(w_{t+j}|w_t) \qquad (1)$$

where $c$ is the size of the training context (which can be a function of the center word $w_t$). Larger $c$ results in more training examples and thus can lead to a higher accuracy, at the expense of the

training time. The basic Skip-gram formulation defines $p(w_{t+j}|w_t)$ using the softmax function:

$$p(w_O|w_I) = \frac{\exp\left({v'_{w_O}}^\top v_{w_I}\right)}{\sum_{w=1}^{W} \exp\left({v'_w}^\top v_{w_I}\right)} \tag{2}$$

where $v_w$ and $v'_w$ are the "input" and "output" vector representations of $w$, and $W$ is the number of words in the vocabulary. This formulation is impractical because the cost of computing $\nabla \log p(w_O|w_I)$ is proportional to $W$, which is often large ($10^5$–$10^7$ terms).

## 2.1 Hierarchical Softmax

A computationally efficient approximation of the full softmax is the hierarchical softmax. In the context of neural network language models, it was first introduced by Morin and Bengio [12]. The main advantage is that instead of evaluating $W$ output nodes in the neural network to obtain the probability distribution, it is needed to evaluate only about $\log_2(W)$ nodes.

The hierarchical softmax uses a binary tree representation of the output layer with the $W$ words as its leaves and, for each node, explicitly represents the relative probabilities of its child nodes. These define a random walk that assigns probabilities to words.

More precisely, each word $w$ can be reached by an appropriate path from the root of the tree. Let $n(w, j)$ be the $j$-th node on the path from the root to $w$, and let $L(w)$ be the length of this path, so $n(w, 1) = \text{root}$ and $n(w, L(w)) = w$. In addition, for any inner node $n$, let $\text{ch}(n)$ be an arbitrary fixed child of $n$ and let $[\![x]\!]$ be 1 if $x$ is true and -1 otherwise. Then the hierarchical softmax defines $p(w_O|w_I)$ as follows:

$$p(w|w_I) = \prod_{j=1}^{L(w)-1} \sigma\left([\![n(w, j+1) = \text{ch}(n(w, j))]\!] \cdot {v'_{n(w,j)}}^\top v_{w_I}\right) \tag{3}$$

where $\sigma(x) = 1/(1 + \exp(-x))$. It can be verified that $\sum_{w=1}^{W} p(w|w_I) = 1$. This implies that the cost of computing $\log p(w_O|w_I)$ and $\nabla \log p(w_O|w_I)$ is proportional to $L(w_O)$, which on average is no greater than $\log W$. Also, unlike the standard softmax formulation of the Skip-gram which assigns two representations $v_w$ and $v'_w$ to each word $w$, the hierarchical softmax formulation has one representation $v_w$ for each word $w$ and one representation $v'_n$ for every inner node $n$ of the binary tree.

The structure of the tree used by the hierarchical softmax has a considerable effect on the performance. Mnih and Hinton explored a number of methods for constructing the tree structure and the effect on both the training time and the resulting model accuracy [10]. In our work we use a binary Huffman tree, as it assigns short codes to the frequent words which results in fast training. It has been observed before that grouping words together by their frequency works well as a very simple speedup technique for the neural network based language models [5, 8].

## 2.2 Negative Sampling

An alternative to the hierarchical softmax is Noise Contrastive Estimation (NCE), which was introduced by Gutmann and Hyvarinen [4] and applied to language modeling by Mnih and Teh [11]. NCE posits that a good model should be able to differentiate data from noise by means of logistic regression. This is similar to hinge loss used by Collobert and Weston [2] who trained the models by ranking the data above noise.

While NCE can be shown to approximately maximize the log probability of the softmax, the Skip-gram model is only concerned with learning high-quality vector representations, so we are free to simplify NCE as long as the vector representations retain their quality. We define Negative sampling (NEG) by the objective

$$\log \sigma({v'_{w_O}}^\top v_{w_I}) + \sum_{i=1}^{k} \mathbb{E}_{w_i \sim P_n(w)}\left[\log \sigma(-{v'_{w_i}}^\top v_{w_I})\right] \tag{4}$$

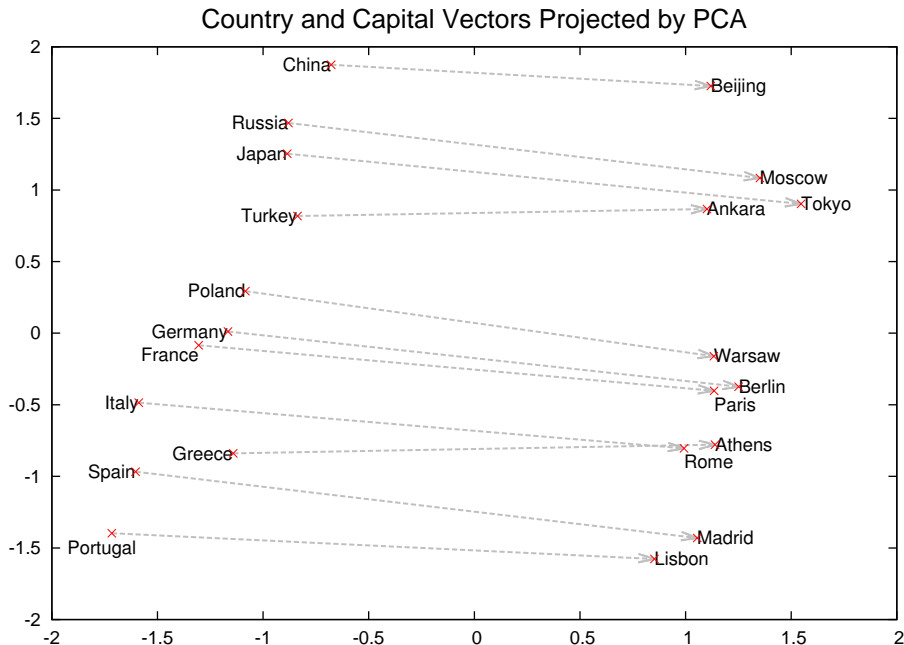

Figure 2: Two-dimensional PCA projection of the 1000-dimensional Skip-gram vectors of countries and their capital cities. The figure illustrates ability of the model to automatically organize concepts and learn implicitly the relationships between them, as during the training we did not provide any supervised information about what a capital city means.

which is used to replace every $\log P(w_O|w_I)$ term in the Skip-gram objective. Thus the task is to distinguish the target word $w_O$ from draws from the noise distribution $P_n(w)$ using logistic regression, where there are $k$ negative samples for each data sample. Our experiments indicate that values of $k$ in the range 5–20 are useful for small training datasets, while for large datasets the $k$ can be as small as 2–5. The main difference between the Negative sampling and NCE is that NCE needs both samples and the numerical probabilities of the noise distribution, while Negative sampling uses only samples. And while NCE approximately maximizes the log probability of the softmax, this property is not important for our application.

Both NCE and NEG have the noise distribution $P_n(w)$ as a free parameter. We investigated a number of choices for $P_n(w)$ and found that the unigram distribution $U(w)$ raised to the $3/4$rd power (i.e., $U(w)^{3/4}/Z$) outperformed significantly the unigram and the uniform distributions, for both NCE and NEG on every task we tried including language modeling (not reported here).

### 2.3 Subsampling of Frequent Words

In very large corpora, the most frequent words can easily occur hundreds of millions of times (e.g., "in", "the", and "a"). Such words usually provide less information value than the rare words. For example, while the Skip-gram model benefits from observing the co-occurrences of "France" and "Paris", it benefits much less from observing the frequent co-occurrences of "France" and "the", as nearly every word co-occurs frequently within a sentence with "the". This idea can also be applied in the opposite direction; the vector representations of frequent words do not change significantly after training on several million examples.

To counter the imbalance between the rare and frequent words, we used a simple subsampling approach: each word $w_i$ in the training set is discarded with probability computed by the formula

$$P(w_i) = 1 - \sqrt{\frac{t}{f(w_i)}} \tag{5}$$

| Method | Time [min] | Syntactic [%] | Semantic [%] | Total accuracy [%] |
|---|---|---|---|---|
| NEG-5 | 38 | 63 | 54 | 59 |
| NEG-15 | 97 | 63 | 58 | **61** |
| HS-Huffman | 41 | 53 | 40 | 47 |
| NCE-5 | 38 | 60 | 45 | 53 |
| The following results use $10^{-5}$ subsampling | | | | |
| NEG-5 | 14 | 61 | 58 | 60 |
| NEG-15 | 36 | 61 | 61 | **61** |
| HS-Huffman | 21 | 52 | 59 | 55 |

Table 1: Accuracy of various Skip-gram 300-dimensional models on the analogical reasoning task as defined in [8]. NEG-$k$ stands for Negative Sampling with $k$ negative samples for each positive sample; NCE stands for Noise Contrastive Estimation and HS-Huffman stands for the Hierarchical Softmax with the frequency-based Huffman codes.

where $f(w_i)$ is the frequency of word $w_i$ and $t$ is a chosen threshold, typically around $10^{-5}$. We chose this subsampling formula because it aggressively subsamples words whose frequency is greater than $t$ while preserving the ranking of the frequencies. Although this subsampling formula was chosen heuristically, we found it to work well in practice. It accelerates learning and even significantly improves the accuracy of the learned vectors of the rare words, as will be shown in the following sections.

## 3  Empirical Results

In this section we evaluate the Hierarchical Softmax (HS), Noise Contrastive Estimation, Negative Sampling, and subsampling of the training words. We used the analogical reasoning task[1] introduced by Mikolov et al. [8]. The task consists of analogies such as "Germany" : "Berlin" :: "France" : ?, which are solved by finding a vector **x** such that vec(**x**) is closest to vec("Berlin") - vec("Germany") + vec("France") according to the cosine distance (we discard the input words from the search). This specific example is considered to have been answered correctly if **x** is "Paris". The task has two broad categories: the syntactic analogies (such as "quick" : "quickly" :: "slow" : "slowly") and the semantic analogies, such as the country to capital city relationship.

For training the Skip-gram models, we have used a large dataset consisting of various news articles (an internal Google dataset with one billion words). We discarded from the vocabulary all words that occurred less than 5 times in the training data, which resulted in a vocabulary of size 692K. The performance of various Skip-gram models on the word analogy test set is reported in Table 1. The table shows that Negative Sampling outperforms the Hierarchical Softmax on the analogical reasoning task, and has even slightly better performance than the Noise Contrastive Estimation. The subsampling of the frequent words improves the training speed several times and makes the word representations significantly more accurate.

It can be argued that the linearity of the skip-gram model makes its vectors more suitable for such linear analogical reasoning, but the results of Mikolov et al. [8] also show that the vectors learned by the standard sigmoidal recurrent neural networks (which are highly non-linear) improve on this task significantly as the amount of the training data increases, suggesting that non-linear models also have a preference for a linear structure of the word representations.

## 4  Learning Phrases

As discussed earlier, many phrases have a meaning that is not a simple composition of the meanings of its individual words. To learn vector representation for phrases, we first find words that appear frequently together, and infrequently in other contexts. For example, "New York Times" and "Toronto Maple Leafs" are replaced by unique tokens in the training data, while a bigram "this is" will remain unchanged.

| Newspapers | | | |
|---|---|---|---|
| New York | New York Times | Baltimore | Baltimore Sun |
| San Jose | San Jose Mercury News | Cincinnati | Cincinnati Enquirer |
| NHL Teams | | | |
| Boston | Boston Bruins | Montreal | Montreal Canadiens |
| Phoenix | Phoenix Coyotes | Nashville | Nashville Predators |
| NBA Teams | | | |
| Detroit | Detroit Pistons | Toronto | Toronto Raptors |
| Oakland | Golden State Warriors | Memphis | Memphis Grizzlies |
| Airlines | | | |
| Austria | Austrian Airlines | Spain | Spainair |
| Belgium | Brussels Airlines | Greece | Aegean Airlines |
| Company executives | | | |
| Steve Ballmer | Microsoft | Larry Page | Google |
| Samuel J. Palmisano | IBM | Werner Vogels | Amazon |

Table 2: Examples of the analogical reasoning task for phrases (the full test set has 3218 examples). The goal is to compute the fourth phrase using the first three. Our best model achieved an accuracy of 72% on this dataset.

This way, we can form many reasonable phrases without greatly increasing the size of the vocabulary; in theory, we can train the Skip-gram model using all n-grams, but that would be too memory intensive. Many techniques have been previously developed to identify phrases in the text; however, it is out of scope of our work to compare them. We decided to use a simple data-driven approach, where phrases are formed based on the unigram and bigram counts, using

$$\text{score}(w_i, w_j) = \frac{\text{count}(w_i w_j) - \delta}{\text{count}(w_i) \times \text{count}(w_j)}. \tag{6}$$

The $\delta$ is used as a discounting coefficient and prevents too many phrases consisting of very infrequent words to be formed. The bigrams with score above the chosen threshold are then used as phrases. Typically, we run 2-4 passes over the training data with decreasing threshold value, allowing longer phrases that consists of several words to be formed. We evaluate the quality of the phrase representations using a new analogical reasoning task that involves phrases. Table 2 shows examples of the five categories of analogies used in this task. This dataset is publicly available on the web[2].

## 4.1 Phrase Skip-Gram Results

Starting with the same news data as in the previous experiments, we first constructed the phrase based training corpus and then we trained several Skip-gram models using different hyperparameters. As before, we used vector dimensionality 300 and context size 5. This setting already achieves good performance on the phrase dataset, and allowed us to quickly compare the Negative Sampling and the Hierarchical Softmax, both with and without subsampling of the frequent tokens. The results are summarized in Table 3.

The results show that while Negative Sampling achieves a respectable accuracy even with $k = 5$, using $k = 15$ achieves considerably better performance. Surprisingly, while we found the Hierarchical Softmax to achieve lower performance when trained without subsampling, it became the best performing method when we downsampled the frequent words. This shows that the subsampling can result in faster training and can also improve accuracy, at least in some cases.

| Method | Dimensionality | No subsampling [%] | $10^{-5}$ subsampling [%] |
|---|---|---|---|
| NEG-5 | 300 | 24 | 27 |
| NEG-15 | 300 | 27 | 42 |
| HS-Huffman | 300 | 19 | **47** |

Table 3: Accuracies of the Skip-gram models on the phrase analogy dataset. The models were trained on approximately one billion words from the news dataset.

|  | NEG-15 with $10^{-5}$ subsampling | HS with $10^{-5}$ subsampling |
|---|---|---|
| Vasco de Gama | Lingsugur | Italian explorer |
| Lake Baikal | Great Rift Valley | Aral Sea |
| Alan Bean | Rebbeca Naomi | moonwalker |
| Ionian Sea | Ruegen | Ionian Islands |
| chess master | chess grandmaster | Garry Kasparov |

Table 4: Examples of the closest entities to the given short phrases, using two different models.

| Czech + currency | Vietnam + capital | German + airlines | Russian + river | French + actress |
|---|---|---|---|---|
| koruna | Hanoi | airline Lufthansa | Moscow | Juliette Binoche |
| Check crown | Ho Chi Minh City | carrier Lufthansa | Volga River | Vanessa Paradis |
| Polish zolty | Viet Nam | flag carrier Lufthansa | upriver | Charlotte Gainsbourg |
| CTK | Vietnamese | Lufthansa | Russia | Cecile De |

Table 5: Vector compositionality using element-wise addition. Four closest tokens to the sum of two vectors are shown, using the best Skip-gram model.

To maximize the accuracy on the phrase analogy task, we increased the amount of the training data by using a dataset with about 33 billion words. We used the hierarchical softmax, dimensionality of 1000, and the entire sentence for the context. This resulted in a model that reached an accuracy of **72%**. We achieved lower accuracy 66% when we reduced the size of the training dataset to 6B words, which suggests that the large amount of the training data is crucial.

To gain further insight into how different the representations learned by different models are, we did inspect manually the nearest neighbours of infrequent phrases using various models. In Table 4, we show a sample of such comparison. Consistently with the previous results, it seems that the best representations of phrases are learned by a model with the hierarchical softmax and subsampling.

## 5 Additive Compositionality

We demonstrated that the word and phrase representations learned by the Skip-gram model exhibit a linear structure that makes it possible to perform precise analogical reasoning using simple vector arithmetics. Interestingly, we found that the Skip-gram representations exhibit another kind of linear structure that makes it possible to meaningfully combine words by an element-wise addition of their vector representations. This phenomenon is illustrated in Table 5.

The additive property of the vectors can be explained by inspecting the training objective. The word vectors are in a linear relationship with the inputs to the softmax nonlinearity. As the word vectors are trained to predict the surrounding words in the sentence, the vectors can be seen as representing the distribution of the context in which a word appears. These values are related logarithmically to the probabilities computed by the output layer, so the sum of two word vectors is related to the product of the two context distributions. The product works here as the AND function: words that are assigned high probabilities by both word vectors will have high probability, and the other words will have low probability. Thus, if "Volga River" appears frequently in the same sentence together with the words "Russian" and "river", the sum of these two word vectors will result in such a feature vector that is close to the vector of "Volga River".

## 6 Comparison to Published Word Representations

Many authors who previously worked on the neural network based representations of words have published their resulting models for further use and comparison: amongst the most well known authors are Collobert and Weston [2], Turian et al. [17], and Mnih and Hinton [10]. We downloaded their word vectors from the web[3]. Mikolov et al. [8] have already evaluated these word representations on the word analogy task, where the Skip-gram models achieved the best performance with a huge margin.

| Model (training time) | Redmond | Havel | ninjutsu | graffiti | capitulate |
|---|---|---|---|---|---|
| Collobert (50d) (2 months) | conyers lubbock keene | plauen dzerzhinsky osterreich | reiki kohona karate | cheesecake gossip dioramas | abdicate accede rearm |
| Turian (200d) (few weeks) | McCarthy Alston Cousins | Jewell Arzu Ovitz | - - - | gunfire emotion impunity | - - - |
| Mnih (100d) (7 days) | Podhurst Harlang Agarwal | Pontiff Pinochet Rodionov | - - - | anaesthetics monkeys Jews | Mavericks planning hesitated |
| Skip-Phrase (1000d, 1 day) | Redmond Wash. Redmond Washington Microsoft | Vaclav Havel president Vaclav Havel Velvet Revolution | ninja martial arts swordsmanship | spray paint grafitti taggers | capitulation capitulated capitulating |

Table 6: Examples of the closest tokens given various well known models and the Skip-gram model trained on phrases using over 30 billion training words. An empty cell means that the word was not in the vocabulary.

To give more insight into the difference of the quality of the learned vectors, we provide empirical comparison by showing the nearest neighbours of infrequent words in Table 6. These examples show that the big Skip-gram model trained on a large corpus visibly outperforms all the other models in the quality of the learned representations. This can be attributed in part to the fact that this model has been trained on about 30 billion words, which is about two to three orders of magnitude more data than the typical size used in the prior work. Interestingly, although the training set is much larger, the training time of the Skip-gram model is just a fraction of the time complexity required by the previous model architectures.

# 7 Conclusion

This work has several key contributions. We show how to train distributed representations of words and phrases with the Skip-gram model and demonstrate that these representations exhibit linear structure that makes precise analogical reasoning possible. The techniques introduced in this paper can be used also for training the continuous bag-of-words model introduced in [8].

We successfully trained models on several orders of magnitude more data than the previously published models, thanks to the computationally efficient model architecture. This results in a great improvement in the quality of the learned word and phrase representations, especially for the rare entities. We also found that the subsampling of the frequent words results in both faster training and significantly better representations of uncommon words. Another contribution of our paper is the Negative sampling algorithm, which is an extremely simple training method that learns accurate representations especially for frequent words.

The choice of the training algorithm and the hyper-parameter selection is a task specific decision, as we found that different problems have different optimal hyperparameter configurations. In our experiments, the most crucial decisions that affect the performance are the choice of the model architecture, the size of the vectors, the subsampling rate, and the size of the training window.

A very interesting result of this work is that the word vectors can be somewhat meaningfully combined using just simple vector addition. Another approach for learning representations of phrases presented in this paper is to simply represent the phrases with a single token. Combination of these two approaches gives a powerful yet simple way how to represent longer pieces of text, while having minimal computational complexity. Our work can thus be seen as complementary to the existing approach that attempts to represent phrases using recursive matrix-vector operations [16].

We made the code for training the word and phrase vectors based on the techniques described in this paper available as an open-source project[4].

## Footnotes

[1]`code.google.com/p/word2vec/source/browse/trunk/questions-words.txt`

[2]code.google.com/p/word2vec/source/browse/trunk/questions-phrases.txt

[3] http://metaoptimize.com/projects/wordreprs/

[4]`code.google.com/p/word2vec`

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
