[Reviews · NeurIPS 2013]

Submitted by Assigned_Reviewer_3

This paper has 3 main contributions: (i) it extends the skip-gram model of (Mikolov et al, 2013) to speed up training by adopting an objective close to Noise Contrastive Estimation and subsampling frequent words, (ii) it proposes to learn phrase representations and introduces a dataset to evaluate those, (iii) it introduces the concept of additive compositionality.

This is a good paper. It is clear and reads well. It is technically correct. The part on speeding up learning (i) justifies the paper. The additive compositionality (iii) is a nice concept to throw in, even if the authors make no attempt to explain how it arises from the optimization problem that training is solving. The part on phrase representation (ii) could be given more emphasis given it could be more controversial. In particular, testing the limits of the proposed technique might make the paper more insightful.

** NCE & Subsampling **

I feel that you should spend more time introducing NCE and highlighting the difference with ICE. From the text, it seems that the notation in (4) is wrong: you sample k sample w according to Pn(w) then you use each individual sample log(sigma<,>) as an unbiased estimator of the expectation but you never manipulate the expectation itself.

For subsampling frequent words, it might be worth mentioning that this strategy might be inefficient for less semantic NLP task (e.g. POS tagging or parsing) where the representation of common words might matter more than for the semantic test from [7].

** Additive compositionality **

This property is similar to the one highlighted in [Tomas Mikolov, Wen-tau Yih, and Geoffrey Zweig, Linguistic Regularities in Continuous SpaceWord Representations] which observed
[Koruna-Czech] \simeq [Yen-Japan]
You make the remark that these two kind of vectors are also close to currency. This is an interesting observation. It would be nice if one would attempt to explain how the optimization problem solved by maximizing (4) actually yields to such property. This remains puzzling to me and maybe to some other member of the ML community.

** Phrase representation **

You propose to learn a representation for each frequent (according to (6)) phrase. This is a valid proposal and your "Air Canada" example against Socher like models make sense. I do not have a strong argument against your strategy when data are plentiful. However, a strong argument in favor of distributed LM as opposed to n-gram LMs that they allow parameter sharing across similar contexts. This help modeling language because there will always be infrequent phrases that can benefit from similar frequent phrases, e.g. learning that "the (small town name) warriors" is a sports team need to share parameters with other popular teams. It might worse testing the limit of your strategy in terms of number of occurrences as it might go a long way.

** Typos **

- Table 3: samsampling -> subsampling
- It might be worth citing Tomas Mikolov, Wen-tau Yih, and Geoffrey Zweig, Linguistic Regularities in Continuous SpaceWord Representations which first introduced the unigram fact dataset.
Summary: This is a good paper. It is clear and reads well. It is technically correct. The part on speeding up learning (i) justifies the paper. The additive compositionality (iii) is a nice concept to throw in, even if the authors make no attempt to explain how it arises from the optimization problem that training is solving. The part on phrase representation (ii) could be given more emphasis given it could be more controversial. In particular, testing the limits of the proposed technique might make the paper more insightful.

Submitted by Assigned_Reviewer_6

This paper proposes 3 improvements for the skip-gram model which allows for learning embeddings for words. The first improvement is subsampling frequent word, the second is the use of a simplified version of noise constrastive estimation (NCE) and finally they propose a method to learn idiomatic phrase embeddings. In all three cases the improvements are somewhat ad-hoc. In practice, both the subsampling and negative samples help to improve generalization substantially on an analogical reasoning task. The paper reviews related work and furthers the interesting topic of additive compositionality in embeddings.

The article does not propose any explanation as to why the negative sampling produces better results than NCE which it is suppose to loosely approximate. In fact it doesn't explain why besides the obvious generalization gain the negative sampling scheme should be preferred to NCE since they achieve similar speeds.

Table 6 is a misrepresentation of past work and should be improved before submission. There are two problems:
1. The paper claims the skip-gram model learns better embeddings based on this evaluation. However, the training set for the skip-gram model is as much as 30x bigger. To make this claim the models would need to be trained on the same training set. As it stands, it can only be said that as ready-made embeddings they are better.
2. The table does not acknowledge the fact that these are rare words (due to random selection). If the aim is to compare the performance on rare words, this needs to be said explicitly. If not, simply randomly select words according to their unigram probability.
Summary: The paper describes improvements over the skip-gram model. The improvements lead to better generalization error in the experiments. However, the comparison to other embedding methods in Table 6 can be misleading and should be addressed.

Submitted by Assigned_Reviewer_7

Summary:
The paper discusses a number of extensions to the Skip-gram model previously proposed by Mikolov et al (citation [7] in the paper): which learns linear word embeddings that are particularly useful for analogical reasoning type tasks. The extensions proposed (namely, negative sampling and sub-sampling of high frequency words) enable extremely fast training of the model on large scale datasets. This also results in significantly improved performance as compared to previously proposed techniques based on neural networks. The authors also provide a method for training phrase level embeddings by slightly tweaking the original training algorithm.
The various proposed extensions to Skip-gram algorithm are compared with respect to speed and accuracy on "some" large scale dataset. The authors also qualitatively compare the goodness of the learnt embeddings with those provided in the literature using neural network models.

Quality:
I found the quality of research discussed in the paper to be above average. The extensions proposed in the paper does result in significant improvements in training of the original skip-gram model. However there are a number of issues in the paper which are worth pointing out.

-- The authors only give a qualitative comparison of the goodness of the embeddings learnt by their model versus those proposed by others in the literature. Any reason why? Ideally, one should compare the performance of learnt embeddings quantitatively by using them on potentially multiple NLP tasks. In the end, no matter how good the embeddings look to the human eye, they are not useful if they cannot be used in a concrete NLP task.

-- Also, I though the comparison (Table 6) is a bit unfair, because the previous embeddings are trained on a different and smaller dataset as compared to the ones learnt in the paper. A more fair result would be to learn all the embeddings on the same dataset.

Clarity:
For the most part the paper is clearly written. There are a number of areas where it needs improvement though. For instance,

-- the authors need to elaborate on how the various hyper-parameters in the extensions proposed are chosen.

-- The details of the dataset used by the authors is completely missing. I have no idea about the nature and source of the data being used for training the models.

Originality:
The paper gives a number of incremental extensions to the original Skip-gram algorithm for learning word embeddings. Though these extensions seem to perform well, I would not call the contributions significantly original.

Significance:
I think the results in the paper are fairly interesting and certainly require further research. I was particularly impressed by how cleanly the model was able to learn embeddings associated with phrases composed of multiple words. Equally interesting was how one could combine (add/subtract) word vectors to generate new meaningful vectors. These properties of the proposed model are certainly worth further attention towards achieving the goal of creating meaningful vector representation for full sentences.


Summary: The paper gives a number of interesting extensions to the original Skip-gram model for training word embeddings in a linear fashion. The work presented is significant enough to be accepted in NIPS and shared among other researchers in the area.
Author Feedback

Author rebuttal: We thank the reviewers for their detailed comments.

R6 + R7:

We will improve our comparison to previous work and explicitly state
that we used more data in the experiments. The aim of Table 6 was
not to argue that the Skip-gram model learns better representation on
the same amount of data. Instead, we tried to show that, if our goal is to learn the best possible
word representations (perhaps because we have an application in mind),
then the Skip-gram model is the best choice precisely because it is much
so much faster than other methods and can therefore be trained on much more data.

As for the words in Table 6 being rare, we deliberately chose them to be so (see line 373), because a common
objection to word vectors is that they are great for the common words
but not useful for rare words.


R7:

We agree that the ultimate test for the quality of word embeddings is their usefulness for
other NLP tasks, and we have already demonstrated that it is the case for machine translation
in a followup submission. However, it is also the case that the analogical
reasoning tasks we use to evaluate the word vectors are at least a
somewhat reasonable metric of word vector quality.

We will release the code to make our results reproducible.